# How to be a Good Teacher?
# Process Strong Pretrained Models for Effective Knowledge Distillation

## Abstract

Transferring the world knowledge encoded in pretrained models through knowledge distillation is an effective approach to improve the performance of small, task-specific production models. However, the effectiveness of such knowledge transfer drops greatly for strong models that are pretrained in a large scale. In this paper, we explore methods to preprocess strong pretrained models to improve the effectiveness of its knowledge transfer. From a mutual information perspective of distillation effectiveness, we propose to incorporate mutual information-aware optimization into the fine-tuning of strong pretrained models. For small or highly-imbalanced downstream datasets where such optimization is less effective, we further propose to heuristically reweight the MLP blocks, which is inspired by our observation that top MLP blocks often cause the loss of mutual information. Our method enables small student models to benefit from those pretrained models among the strongest.

## 1 Introduction

Large-size models pretrained on large-scale general-domain data have achieved great successes in many real-world applications. These models are believed to encode world knowledge (Peng et al., 2023) and are able to generalize to specific tasks with proper adaptation. Nevertheless, the shear size of these models brings about significant amount of serving cost, which impedes their product-level deployment. A straightforward way to utilize these strong pretrained models without inducing additional serving cost is through knowledge distillation, where one hopes that the encoded world knowledge that is relevant to the specific task can be effectively transferred to existing small models, typically with a much smaller size.

The off-the-shelf paradigms to distill pretrained models on specific downstream tasks often consist of two stages. First, the pretrained model is adapted to the downstream task through fine-tuning, where the pretrained model is specialized and competitive performance can be achieved. Second, the fine-tuned pretrained model is distilled into desired product-level small models, through matching the predictions or intermediate features between two models.

However, such a paradigm may not be ideal as existing results have shown that knowledge distillation may be less effective on strong models (Wang et al., 2022), either when model size is scaled up (Cho & Hariharan, 2019) or when advanced training strategies are employed (Müller et al., 2019). Indeed, we observed that distilling strong pretrained models results in non-competitive small models, often not better than those distilled from much weaker pretrained models. To mitigate this issue, a wide spectrum of knowledge distillation algorithms have been proposed, for example, by employing mid-size models as assistants to bridge the capacity gap (Mirzadeh et al., 2019), by matching the intra-class relation of the predictions (Huang et al., 2022), or by pruning the distillation signals from those difficult classes (Zhu et al., 2022).

In this paper, we argue that effective knowledge transfer from strong pretrained models requires not only advanced distillation algorithms, but also proper preprocessing of these pretrained models before distillation. After all, the problem that a student cannot learn well from the teacher may not entirely indicate that the student's learning strategy is ineffective, but more likely suggests that the teacher is not properly aligned for instruction.

To tackle this problem, we motivate from the existing understanding of the effectiveness of knowledge distillation, where it has been empirically observed (Müller et al., 2019) and theoretically argued (Wang et al., 2022) that the mutual information between the input data and the distillation targets is critical for knowledge distillation. We observed that the relative ineffectiveness of knowledge transfer from strong pretrained models can also be attributed to the low mutual information. Inspired by this, we explore simple and efficient methods to improve the mutual information during fine-tuning. We found that Sharpness-Aware Minimization (SAM) (Foret et al., 2020), an optimization technique widely used to improve model generalization, if properly tuned, can be employed to effectively improve mutual information and the distillation effectiveness as well.

When the downstream dataset is small, the above mutual information-aware optimization may not be effective. We thus explore methods that directly process the model modules to improve the mutual information. We focus on the Transformer architecture. By gradually pruning MLP and self-attention blocks from top to bottom, we found that the depletion of mutual information in strong pretrained models can be largely attributed to top MLP blocks. We further found that top MLP blocks deplete mutual information is mainly the result of their high *expertness*, which roughly measures how sparse the MLP block's neuron activations are for a subset of similar input examples. We prove that high expertness of an MLP block naturally bottlenecks its mutual information. As an extreme case for an intuition, when a subset of inputs only activate a single neuron, the block outputs converge to the same representation, which obscures any information within this set of inputs. Based on this observation, we propose to simply downweight the top MLP blocks of strong pretrained models to improve the mutual information.

We combine both SAM and our simple block reweighting heuristics to improve the effectiveness of knowledge distillation for strong pretrained models. On a variety of tasks, model architectures, and fine-tuning methods, we observe consistent gain of the small student model's performance upon distillation. Furthermore, when scaling up the size of the pretrained model and the scale of the pretraining data, we observe that the small student model continues to benefit from the increasing performance of the strong teacher models.

## 2 RELATED WORK

**Understand the difficulty of distilling strong models.** Existing works have shown that strong models as knowledge distillation teachers may be less effective, or even consistently hurt the small model's performance compared to weak models (Cho & Hariharan, 2019; Huang et al., 2022). Specifically, early-stopped checkpoints of the teacher model, or even the snapshot ensemble (Huang et al., 2017) of these checkpoints, may be beneficial to the student performance (Wang et al., 2022). Dao et al. (2021) proved that strong teacher can overfit the training set, thus may deviate its probabilistic predictions from the Bayes class probabilities of the data distribution. Zhu et al. (2022) found that, compared to vanilla training, distilling from strong teachers may hurt the student's performance on certain classes, which they refer to as "undistillable classes".

**Student-oriented teacher training.** The main focus of this paper is to distill strong pretrained models on specific downstream tasks. Distillation in this scenario can be particularly challenging, potentially due to not only the gap between teacher-student model capacity, but also the gap between the pretrained and downstream datasets. We are thus particularly interested in how to process the pretrained model to improve its distillation effectiveness.

Existing works have been focusing on training teachers for better student performance, albeit in the standard full training setup instead of pretraining-fine-tuning setup. Dao et al. (2021) proposes to conduct cross-fitting when training the teacher. Specifically, the training data is split into several folds, where the teacher predictions on each fold are generated by the model trained only on out-of-fold data. Park et al. (2021) jointly trains the teacher model and student's model blocks, aiming to impose regularizations toward the student performance. Dong et al. (2022) explores the necessary conditions for the teacher model to learn Bayes probabilistic predictions and proposes to incorporate additional regularization into the teacher training, which is shown to improve the student performance. For multi-generalization distillation specifically, Yang et al. (2019) proposes to penalize the difference between the probabilistic prediction at the true class and these at other semantically relevant classes to encourage the learning of secondary probabilities in teacher training, which can improve the performance of models in later generations. However, these methods may not be readily applied to

the distillation of large pretrained models, either due to significant computation overhead, or strict constraints on the teacher's model architecture.

# 3 MUTUAL INFORMATION-AWARE FINE-TUNING OF STRONG PRETRAINED MODELS

**Understand distillation effectiveness from a mutual information perspective.** As some necessary context, standard knowledge distillation jointly optimizes two loss functions in a multi-class classification problem, including the cross-entropy loss with the given hard labels $Y$, and the soft labels or intermediate features of the teacher, namely $L_{\text{KD}} = \lambda L_{\text{CE}}(Y, P_S) + (1 - \lambda)L(F_T, F_S)$, where $P_S$ is the student logits and $F_T$ and $F_S$ are the logits or intermediate features of the teacher and student respectively. We will refer to $F_T$ as the distillation targets.

Existing works have established rich evidence that the mutual information between the distillation targets and the input data, namely $I(X; F_T) = \mathbb{E}_{X, F_T}[\log(p(F_T|X)) - \log(\mathbb{E}_X p(F_T|X))]$, is critical for knowledge distillation. Intuitively, here mutual information can be understood as the how well one can distinguish the input examples from the distillation targets, which is important for knowledge distillation effectiveness. Müller et al. (2019) shows that in an extreme case where all the information about the input is lost in the distillation targets, they will contain no extra information compared to the hard labels. In such a case, knowledge distillation will be no better than standard training of the student with hard labels. They employ such a principle to explain the ineffectiveness of knowledge distillation when teacher is trained with label smoothing. Wang et al. (2022) further establishes a theory based on information bottleneck (Tishby et al., 2000; Tishby & Zaslavsky, 2015) to show that low mutual information will reduce the effectiveness of knowledge distillation, which they use to explain the advantage of using intermediate training checkpoints for distillation. Therefore, it can be empirically concluded that the effectiveness of knowledge distillation can be modeled by a function of the mutual information with the input data and the teacher's performance, as well as other unknown potential factors, namely

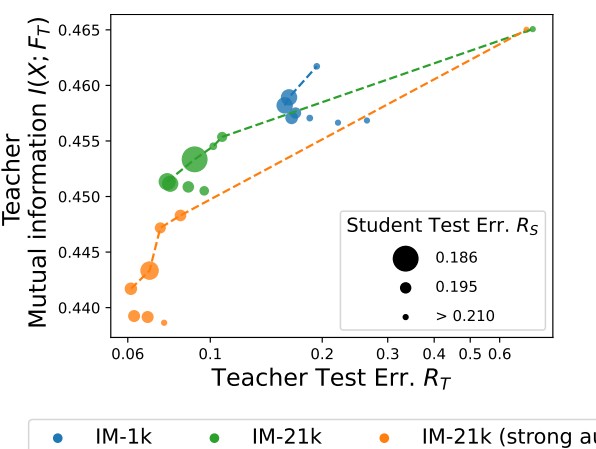

Figure 1: Test error vs. mutual information of pretrained models fine-tuned on different downstream datasets. Colors denote pretrained models of different strengths, including ViT-B models pretrained on ImageNet-1K ("*IM-1k*"), ImageNet-21K ("*IM-21k*"), and ImageNet-21K with strong data augmentations ("*IM-21k (strong aug)*"), ranked by their strengths. Sizes denote the test errors of students distilled from the corresponding fine-tuned models. The downstream dataset is CIFAR-100. For the fine-tuning of each pretrained model, we grid search the learning rate and the number of training steps, and connect the pareto-optimal points (lower test err, higher mutual info) with a dashed line.

$$R_S \sim \gamma(R_T, I(X; F_T), \cdots), \tag{1}$$

where we use the student test error $R_S$ to quantify the effectiveness of the knowledge distillation, and use the teacher test error $R_T$ to denote the teacher's performance. Note that the higher $I(X; F_T)$ and the lower $R_T$ are, the smaller $R_s$ will be. We will refer to the coordinates $(R_T, I(X; F_T))$ as the information plane in the rest of the paper following Wang et al. (2022).

**Understand the difficulty of distilling strong pretrained models.** Following the above principle, we show that the difficulty of distilling strong pretrained models can be attributed to low mutual information. As shown in Figure 1, when the teacher model is pretrained with a large-scale data and

strong data augmentation, though the teacher test error $R_T$ reduces a lot, the mutual information $I(X; F_T)$ [1] also drops significantly, which results in worse student performance.

**Sharpness-aware minimization for a pareto-superior information plane.** Based on Equation 1, to improve the distillation effectiveness of pretrained models, we can process the pretrained model such that it can reach a better pareto-front on the information plane. A straightforward solution here is to incorporate mutual information as an optimization objective in fine-tuning. Standard approaches can utilize backpropable mutual information estimator such as MINE (Belghazi et al., 2018). Nevertheless, we found that Sharpness-Aware Minimization, an optimization technique used to improve model generalization, can be employed to effectively improve mutual information in fine-tuning.

SAM defines the optimization objective as, $\min_W \max_{\|\Delta\|_2 \leq \rho} \frac{1}{N} \sum_{i \in [N]} \ell(x_i, y_i; W + \Delta)$, where $\Delta$ is a perturbation to the model weights $W$ with bounded size $\rho$. Following Foret et al. (2020), we make this minimax problem practical by approximating the inner maximization with a single-step gradient ascent. SAM requires no auxiliary neural network to estimate mutual information and is simple to adapt to existing fine-tuning pipeline.

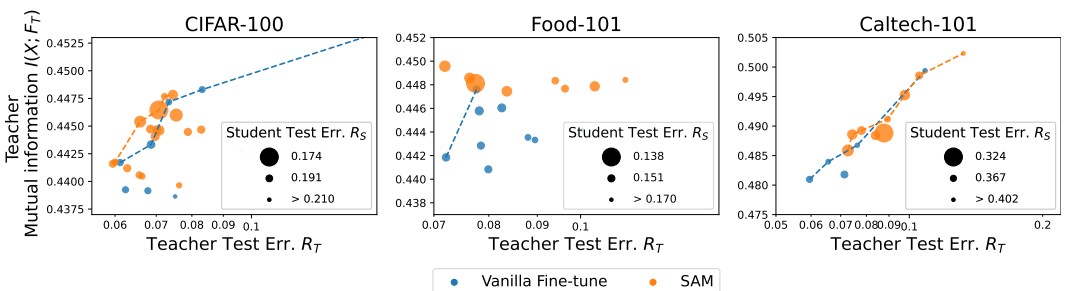

Figure 2: Test error vs. mutual information of pretrained models fine-tuned on different downstream datasets, both in the vanilla way and with SAM. Here the pretrained model is ViT-B pretrained on ImageNet-21k with strong data augmentations. Other setups are similar to Figure 1.

As shown in Figure 2, fine-tuning the pretrained models with SAM can improve mutual information without losing too much performance, and thus reaches a better pareto-front on the information plane. Subsequently, students distilled from these fine-tuned models get improved performance.

Note that here we require a quite large perturbation size ($\rho \gtrsim 0.05$) of SAM to improve mutual information and student performance. Such a value is significantly larger than the typical perturbation size used to improve model generalization with SAM ($\sim$ 0.001), as also shown in Figure 3.

Finally, we notice that using optimization techniques such as SAM to regularize mutual information may not be effective when fine-tuning on small downstream datasets such as Caltech-101, as also shown in Figure 2. To this end, we explore alternative methods that can directly process strong pretrained models towards better mutual information without training.

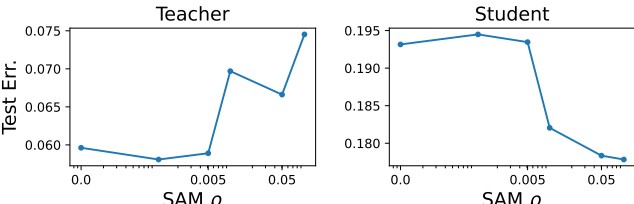

Figure 3: Test error of the pretrained model fine-tuned with SAM of different perturbation sizes $\rho$ (Left), and test error of the student distilled from the corresponding fine-tuned model (Right). $\rho = 0$ denotes vanilla fine-tuning. Here the pretrained model is ViT-B pretrained on ImageNet-21k with strong data augmentations. The downstream dataset is CIFAR-100. We grid search the fine-tuning hyperparameter setup similar to Figure 1 and select the set of hyperparameters that maximizes the fine-tuning performance.

---

[1]We use the reconstruction loss of a decoder to quantify mutual information following (Wang et al., 2021; 2022). See Appendix A.5 for more details.

## 4 HEURISTIC BLOCK REWEIGHTING OF STRONG PRETRAINED MODELS

**Top MLP blocks in strong pretrained models reduce mutual information significantly.** we attempt to localize the low mutual information of strong pretrained models into individual model blocks. Specifically, we gradually prune more blocks in the pretrained model from top to bottom, fine-tune it on the downstream dataset, and estimate the mutual information.

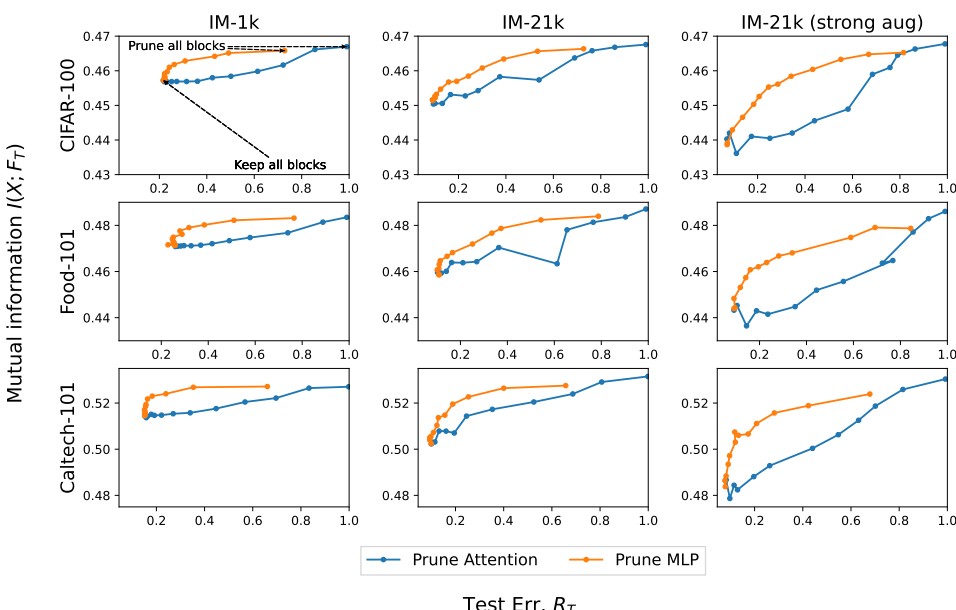

Figure 4: Test error vs. mutual information of pretrained models fine-tuned with more model blocks (MLP or Self-attention) gradually pruned from top to bottom.

As shown by Figure 4, we find that as more self-attention blocks are pruned, the mutual information increases almost linearly as the model performance decreases, which is intuitive. However, as more MLP blocks are pruned, there exists a turning point. The mutual information first increases significantly, then increases moderately with a slope similar to that of pruning the attention blocks. This suggests that top MLP blocks reduce mutual information without improving model performance equally as other blocks. The significance of such reduction becomes more prominent as the pretrained model becomes stronger.

**Reweighting MLP blocks to increase mutual information.** The above observation offers us a simple heuristic to improve the mutual information and thus the distillation effectiveness of strong pretrained models. Specifically, we propose to simply downweight the output of the MLP block in each layer to reduce its negative impact on the mutual information, while keep the attention block intact, namely

$$\tilde{x}_l = x_l + \text{Self-Attention}(x_l),$$
$$x_{l+1} = (2 - \alpha)\tilde{x}_l + \alpha \text{MLP}(\tilde{x}_l), \tag{2}$$

where $x_l$ is the token representation at the $l$-th layer, and $\alpha$ is a hyperparameter to control the weight of the MLP block. $\alpha = 1$ will recover the original pretrained model, while $\alpha < 1$ will downweight the MLP block and upweight the residual connection.

Note that such a reweighting will naturally downweight the top MLP blocks more than the bottom ones. Based on simple arithmetics, one can show that the effective weight $\tilde{\alpha}$ of the MLP block at the $l$-th layer is $\tilde{\alpha}_l = \alpha \cdot (2 - \alpha)^{(l_{\text{tot}} - l)}$, where $l_{\text{tot}}$ is the total number of layers in the transformer model. It can be seen that $\tilde{\alpha}$ drops exponentially for top layers. Our method is very similar to ReZero (Bachlechner et al., 2020), albeit that we also upweight the residual connection in addition to downweight the contribution of each residual block.

## 5 WHY MLP BLOCKS CAUSE LOW MUTUAL INFORMATION?

We take one step further to understand why the top MLP blocks will cause low mutual information in strong pretrained models.

### 5.1 TOP MLP BLOCKS ARE SPONTANEOUS MIXTURE-OF-EXPERTS

It has been widely observed that the neuron activations of MLP blocks are highly sparse (Zhang et al., 2021; Li et al., 2022). Here neuron activations refer to the intermediate outputs of the MLP block after the ReLU activation function. Such sparsity can be as low as only $\sim 1\%$ of the activations are non-zeros for each input to MLP, and is more significant for top MLP blocks and for strong pretrained models (Li et al., 2022). Existing works have successfully utilized such sparsity to convert dense pretrained models to MoE models (Zhang et al., 2021), which consist of MoE MLP blocks in replace of standard MLP blocks. An MoE MLP block is different from a standard MLP block in that it partitions the block parameters into several subsets, which are known as *experts*. Given an input, only one or a few experts will activate, which thus reduces the computation cost. Formally, we define an MoE MLP block as follows [2].

**Definition 5.1** (Notations). We will use $g_\leftarrow(X)$ to denote the input of a MLP block, where $g_\leftarrow$ denotes the prior blocks that map input $X$ to the input of the MLP block. We define $\mathbf{1}_\mathcal{S}$ as the indicator vector of a set of indices $\mathcal{S}$, where the $i$-th entry will be one if $i \in \mathcal{S}$. We define $\odot$ as the element-wise product, $[N] := \{1, 2, \cdots, N\}$, and $|\mathcal{S}|$ as the size of a set $\mathcal{S}$.

**Definition 5.2** (MoE MLP). Let $d_{\text{MLP}}$ be the width of the MLP block (also the total number of neurons) and $d_{\text{embed}}$ be the embedding size. A standard MLP block performs the computation as

$$\text{MLP}(g_\leftarrow(X)) = \phi(g_\leftarrow(X)\boldsymbol{W_1})\boldsymbol{W_2}, \tag{3}$$

where $\boldsymbol{W_1} \in \mathbb{R}^{d_{\text{embed}} \times d_{\text{MLP}}}$ and $\boldsymbol{W_2} \in \mathbb{R}^{d_{\text{MLP}} \times d_{\text{embed}}}$ [3] and $\phi(\cdot)$ is a non-linear activation function. In contrast, an MoE MLP block performs the computation as

$$\text{MLP}(g_\leftarrow(X)) = \mathbf{1}_{\mathcal{S}_Z} \odot \phi(g_\leftarrow(X)\boldsymbol{W_1})\boldsymbol{W_2}, \tag{4}$$

where $Z$ indexes the expert that will be used for input $X$ and $\mathcal{S}_Z \subseteq [d_{\text{MLP}}]$ is the index set of a few neurons that will have non-zero activation on input $X$,

**Measure the "expertness" of a pretrained MLP block.** The success of converting dense MLPs to MoE MLPs implies that expert structures may spontaneously emerge in pretrained transformers. We are interested in to what extent a pretrained dense MLP block resembles a sparse MoE MLP block, which we will refer to as "*expertness*". To define expertness more formally, we view the neuron activations of an MLP block as a bipartite graph, where one subset of nodes represent the input examples to the block, the other subset of nodes represent the neurons of the block, and the edges represent the activation of each neuron on each input. For a very dense MLP block, almost all edges would be non-zero; While for an MoE MLP block, most of the edges would be zero and thus can be pruned. We can now define the expertness of an MLP block as the goodness of the approximation of a maximally pruned graph to the original neuron activation graph.

**Definition 5.3** ("Expertness"). Let $G = (\mathcal{U}, \mathcal{X}, E)$ be a bipartite graph that represents the activations of a set of neurons $\mathcal{U}$ on a set of input examples $\mathcal{X}$, where $E_{ij}$ denote the activation of $i$-th neuron on $j$-th input [4]. We define the cut between two sets of vertices $\mathcal{V}_1$ and $\mathcal{V}_2$ as the total norm of the edges between these two vertices [5], namely $\text{cut}(\mathcal{V}_1, \mathcal{V}_2) = \sum_{i \in \mathcal{V}_1, j \in \mathcal{V}_2} E_{ij}^2$. We also define a bipartition of $G$ as disjoint clusters of the neurons $\{\mathcal{U}_1, \cdots, \mathcal{U}_k\}$ and the corresponding disjoint clusters of the inputs $\{\mathcal{X}_1, \cdots, \mathcal{X}_k\}$, where $\cup_{k'}\mathcal{U}_{k'} = \mathcal{U}$ and $\cup_{k'}\mathcal{X}_{k'} = \mathcal{X}$. Now the expertness $e$ is defined as the maximum total cut that can be achieved by any bipartition, namely

$$e(G) := \frac{1}{\text{cut}(\mathcal{U}, \mathcal{X})} \max_{\{\mathcal{U}_1, \cdots, \mathcal{U}_k\}, \{\mathcal{X}_1, \cdots, \mathcal{X}_k\}} \sum_{k'=1}^{k} \text{cut}(\mathcal{U}_{k'}, \mathcal{X}_{k'}). \tag{5}$$

---

[2] We follow existing works (Fedus et al., 2021) and adopt a typical setting of MoE where only one expert is activated.

[3] We neglected the bias term for simplicity.

[4] We consider the concatenation of all token activations as the activation of an input.

[5] Note that this is also the square of the Frobenius norm of the neuron-input activation matrix.

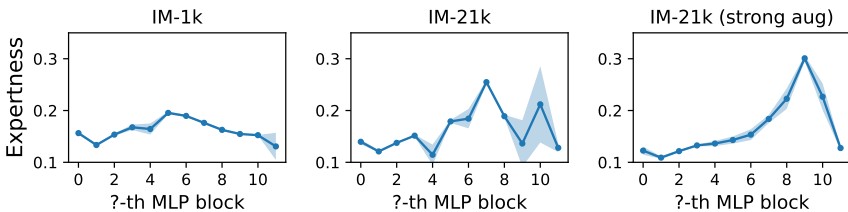

Figure 5: Estimation of the expertness of MLP blocks in pretrained models of various strengths.

We preset the number of experts to be same as the number of classes and use the classic spectral co-clustering algorithm (Dhillon, 2001) to solve this bipartition problem.

**Emergence of "expertness" in top MLP blocks of strong pretrained transformer.** We estimate the expertness of the MLP block at each layer in a pretrained transformer. As shown in Figure 5, top MLP blocks gradually show higher expertness as the pretrained model becomes stronger, which means they tend to activate only a particular subset of the neurons given an input. Note that such expertness emerges spontaneously in pretraining without explicit regularization. Interestingly, the topmost MLP blocks are often not the blocks with the highest expertness. We notice that a similar phenomenon is also observed for the activation sparsity of MLP blocks (Li et al., 2022), which we suspect it might be because the topmost MLP blocks are not sufficiently optimized in pretraining.

## 5.2 HIGH EXPERTNESS CORRELATES WITH LOW MUTUAL INFORMATION

We show that top MLP blocks of strong pretrained models cause low mutual information likely because of their high expertness.

**Intuition of why expertness leads to low mutual information.** The intuition here is that, if the outputs of a sub-population of input examples are contributed by the few neurons in an expert, they will tend to be similar and thus it would be difficult to distinguish the input examples based on these outputs. In this case, these outputs will contain little information in additional to the fact that they are from that particular sub-population. Here the distinguishability of the input examples from outputs can be described by mutual information as mentioned before. And how few the experts would be sufficient to compute the outputs is well measured by expertness.

**An analytical result.** We further provide an analytical result to show expertness of the MLP blocks may cause low mutual information. We consider an MoE MLP block that consists of $M$ experts, where each expert selectively activates a few neurons based on the sub-population that the input example belongs to. We are interested in the mutual information between the output of the MoE MLP and the input $X$, which can be upper bounded as follows.

**Proposition 5.4** (Mutual information of an MoE MLP). *Let $Z$ be a random variable that denotes the index of the expert that will be used for input $X$ and $Z \sim [M]$. Equivalently, $Z$ will also denote the index of the sub-population of $X$ that will share the same expert. We will have*

$$I(MLP(g_\leftarrow(X)); X) \leq I(Z; X) + \sum_{z \in [M]} I(\{\phi(\cdot)_i\}_{i \in \mathcal{S}_z}; X) \leq \log_2 M + \sum_{z \in [M]} |\mathcal{S}_z| b, \quad (6)$$

*where $b$ is the maximum entropy (*i.e.*, bits) of a single neuron activation.*

Here the first "$\leq$" leverages the data processing inequality (DPI) since computation of the MoE MLP is deterministic. The second "$\leq$" comes from the fact that mutual information is bounded by the entropy, namely $I(\cdot; X) \leq H(\cdot)$, and the maximum possible entropy of a discrete random variable $Z$ is $\log_2 M$, while the maximum possible entropy of a neuron activation is related to its precision.

The above proposition shows that in addition to the information about which sub-population that $X$ belongs to, *i.e.*, $I(Z; X)$, the mutual information of the output of the MoE MLP block depletes linearly with the sparsity of the neuron activation in each expert. If the sub-population $Z$ happens to match the class label $Y$, then the only information that the output contains in addition to the class label (which is the exact information that will help distillation since the class label is already available to the student) is bounded by $\sum_{z \in [M]} |\mathcal{S}_z| b$. If the neuron activation is a single-precision float number, *i.e.*, $b = 16$, the size of the class label set is 100, and about 1% of the neurons that are

activated, then the maximum information is about 3000 bits for a ViT-B model ($d_{\text{MLP}} = 3072$). In comparison, a typical input image of size $224 \times 224 \times 3$ contains about $4 \times 10^6$ bits of information.

# 6 EXPERIMENTS

We combine SAM and "memory" block reweighting as our teacher fine-tuning method, which we will refer to as *ReMem*. We experiment with ReMem on a wide variety of downstream tasks, task-specific models, distillation algorithms, and fine-tuning algorithms.

## 6.1 EXPERIMENT SETUP

**Model architectures.** We conduct experiments on distilling strong pretrained models with transformer backbone. The default teacher model will be ViT-Base (Dosovitskiy et al., 2020) pretrained on ImageNet-21k (Kolesnikov et al., 2019), a large-scale dataset for image understanding pretraining. We will use the ViT checkpoints trained with Augreg (Steiner et al., 2021), which are publicly available [6]. By default, we will use the widely used ResNet-18 as the product-level, task-specific model architecture.

**Fine-tuning setup.** For convenience, we rescale the input images from all downstream datasets to $224 \times 224$ for fine-tuning, following (Kornblith et al., 2018). We adopt SGD with momentum as the optimizer. We set weight decay to 0 following (Dehghani et al., 2023). Additional hyperparameter setup can be found in Table 8 in the Appendix, unless otherwise mentioned.

**Distillation setup.** For knowledge distillation algorithms, by default we experiment with the original logit matching method (Hinton et al., 2015) due to its wide applicability irrespective of the teacher and student architecture designs. Detailed hyperparameter setup can be found in Table 8.

**Evaluation.** To *fairly* evaluate a teacher fine-tuning strategy, we always early stop the teacher fine-tuning at multiple checkpoints, distill student from each checkpoint, and select the best student performance across these checkpoints. We will also sweep over other teacher and student hyperparameters, including the learning rates in teacher or student training, and the interpolation weight and temperature in knowledge distillation. We report the best student performance among these different settings for a specific teacher fine-tuning strategy.

Table 1: Test accuracy of the teacher and student with knowledge distillation conducted on various datasets. The student network is ResNet-18, while the teacher network is a ViT-Base pretrained on ImageNet-21k.

| Student | Teacher | Student | Teacher | Student | Teacher | Student | Teacher | Student | Teacher | Student | Teacher | Student | Teacher |
|---------|---------|---------|---------|---------|---------|---------|---------|---------|---------|---------|---------|---------|---------|
| Method | | CIFAR-100 | | Flowers | | Pet | | DTD | | SVHN | | Caltech-101 | |
| Scratch | - | 75.5 | - | 79.6 | - | 59.4 | - | 44.9 | - | 96.4 | - | 51.7 | - |
| Distillation | Fine-tune | 81.2 | 91.1 | 82.6 | 95.5 | 74.8 | 94.0 | 43.2 | 78.0 | **97.4** | 96.9 | 57.6 | 91.0 |
| Distillation | *ReMem* | **83.4** | 92.3 | **92.0** | 99.2 | **83.8** | 91.9 | **58.0** | 70.4 | **97.4** | 97.3 | **77.2** | 86.4 |
| | | Food-101 | | Cars | | SUN397 | | SUN397 (TFDS) | | iNaturalist 2017 | | Patch Camelyon | |
| Scratch | - | 83.4 | - | 84.7 | - | 49.6 | - | 62.8 | - | 42.5 | - | 84.3 | - |
| Distillation | Fine-tune | 85.2 | 93.0 | 86.0 | 89.5 | 56.5 | 77.4 | 65.7 | 76.2 | 41.9 | 63.4 | 91.6 | 91.4 |
| Distillation | *ReMem* | **86.5** | 92.4 | **87.0** | 90.5 | **61.6** | 74.0 | **67.8** | 76.9 | **43.0** | 58.5 | **91.9** | 92.0 |
| | | Retinopathy | | EuroSAT | | Resisc45 | | ImageNet-LT | | | | | |
| Scratch | - | 81.8 | - | 98.7 | - | 93.8 | - | 41.6 | - | | | | |
| Distillation | Fine-tune | **81.9** | 80.9 | 99.2 | 99.1 | 96.2 | 97.3 | 43.7 | 73.2 | | | | |
| Distillation | *ReMem* | 81.8 | 81.7 | **99.3** | 99.1 | **96.4** | 97.2 | **45.2** | 71.8 | | | | |

## 6.2 APPLICABILITY OF REMEM

**Downstream tasks.** We experiment on in total 16 downstream image classification datasets. Most of them are from VTAB (Zhai et al., 2019), a benchmark for visual transfer learning. We select additional datasets from existing transfer learning learning literature (Kornblith et al., 2018) that are easily accessible. We also include relatively large datasets such as iNaturalist. For convenience, we adopt the TensorFlow Datasets [7] train-test splits of these datasets by default.

---

[6] https://github.com/google-research/vision_transformer
[7] https://www.tensorflow.org/datasets

The selected datasets distribute across different specialized domains, such as natural images (e.g., CIFAR-100, Cars), medical images (e.g., Patch Camelyon), and sensing images (e.g., EuroSAT). We include datasets of different sizes, including small datasets (e.g., Flowers, Caltech-101) and medium to large datasets (e.g., SUN397, iNaturalist). We also include datasets that are fine-grained and class-imbalanced (e.g., iNaturalist), or in particular, follow a long-tail distribution in terms of the number of training examples across classes (e.g., ImageNet-LT). Table 7 in the appendix lists the detailed specifications of the selected datasets.

Table 1 shows that our method can consistently improve the student performance across these different datasets. Interestingly, on most datasets, our method in fact reduces the teacher's performance. This implies that our method is dedicated to improving the effectiveness of the knowledge distillation.

**Efficient model architectures.** We experiment on additional task-specific efficient model architectures, including widely used MobileNetV2 (Sandler et al., 2018) and EfficientNetV2 (Tan & Le, 2021). As show in Table 2, our method can consistently improve the performance of these efficient model architectures upon distillation.

Table 2: Performance of the teacher and student with alternative student model architectures. Due to compute constraint, we report results on 6 representative datasets, including CIFAR-100, Caltech-101, SUN397 (TFDS), iNaturalist, Patch Camelyon, and ImageNet-LT. These datasets span over the general and specialized, small and large, course-grained and fine-grained, and class-balanced and long-tailed. The same setup will be applied to Figures 3, 4 as well.

| Student Architecture | Teacher Fine-tuning | CIFAR-100 | | Caltech-101 | | SUN-397 (TFDS) | | iNaturalist | | Patch Camelyon | | ImageNet-LT | |
|---|---|---|---|---|---|---|---|---|---|---|---|---|---|
| | | Student | Teacher | Student | Teacher | Student | Teacher | Student | Teacher | Student | Teacher | Student | Teacher |
| MobileNetV2 | - | 77.8 | 91.0 | 60.7 | 90.8 | 59.5 | 76.2 | 34.8 | 64.9 | 91.0 | 91.3 | 36.1 | 75.7 |
| | *ReMem* | **80.2** | 92.5 | **76.7** | 87.7 | **61.3** | 74.7 | **36.4** | 58.5 | **91.2** | 92.3 | **38.4** | 72.1 |
| EfficientNetV2 | - | 80.0 | 89.1 | 60.0 | 93.4 | 65.3 | 76.2 | 48.5 | 64.0 | 90.0 | 89.9 | 39.7 | 73.2 |
| | *ReMem* | **82.7** | 89.1 | **78.5** | 87.2 | **67.2** | 78.4 | **49.7** | 63.2 | **91.3** | 91.1 | **41.5** | 75.4 |

**Advanced knowledge distillation algorithms.** As the production-level, task-specific models often differ from the pretrained large model significantly in terms of architecture, we experiment on those distillation algorithms that can be readily applicable to dissimilar teacher and student architectures. We thus consider the classic knowledge distillation algorithm (Hinton et al., 2015), also known as logit matching, as well as the DIST algorithm (Huang et al., 2022) that distills the intra-class relation between samples. Recently, it is shown that the classic logit matching algorithm can actually be quite competitive if the student model is trained for a sufficient number of steps (Beyer et al., 2021) and with aggressive data augmentations such as Mixup (Zhang et al., 2017). We refer to this algorithm as patient distillation, and conduct experiments on it as well. As shown in Figure 3, our method can consistently improve the student performance with these different distillation algorithms.

Table 3: Performance of the teacher and student with alternative distillation algorithms.

| Distillation Algorithm | Teacher Fine-tuning | CIFAR-100 | | Caltech-101 | | SUN-397 (TFDS) | | iNaturalist | | Patch Camelyon | | ImageNet-LT | |
|---|---|---|---|---|---|---|---|---|---|---|---|---|---|
| | | Student | Teacher | Student | Teacher | Student | Teacher | Student | Teacher | Student | Teacher | Student | Teacher |
| Patient | - | 82.9 | 94.0 | 77.2 | 95.3 | 64.9 | 74.5 | 41.2 | 65.0 | 91.1 | 91.4 | 44.2 | 76.2 |
| | *ReMem* | **84.4** | 93.0 | **86.8** | 95.0 | **67.1** | 74.8 | **42.6** | 64.7 | **91.7** | 89.7 | **45.4** | 71.9 |
| DIST | - | 80.7 | 93.4 | 57.6 | 90.2 | 65.8 | 77.3 | 39.4 | 64.0 | **91.8** | 91.5 | 40.5 | 75.1 |
| | *ReMem* | **82.9** | 89.3 | **78.1** | 88.5 | **68.0** | 74.6 | **41.6** | 62.9 | 91.2 | 91.8 | **44.3** | 57.2 |

**Efficient fine-tuning methods.** As fine-tuning the large pretrained model can be resource-prohibitive, we also experiment with LoRA (Hu et al., 2021), a widely used parameter-efficient fine-tuning method. As shown in Figure 4, our method can also improve the student performance with this parameter-efficient fine-tuning algorithm.

Table 4: Performance of the teacher and student with alternative parameter-efficient methods for teacher fine-tuning.

| Teacher Fine-tuning | CIFAR-100 | | Caltech-101 | | SUN-397 (TFDS) | | iNaturalist | | Patch Camelyon | | ImageNet-LT | |
|---|---|---|---|---|---|---|---|---|---|---|---|---|
| | Student | Teacher | Student | Teacher | Student | Teacher | Student | Teacher | Student | Teacher | Student | Teacher |
| LoRA | 82.0 | 93.1 | 56.7 | 96.0 | 65.5 | 76.8 | 42.0 | 61.2 | 89.3 | 89.2 | 43.1 | 76.8 |
| LoRA + *ReMem* | **83.4** | 91.6 | **78.8** | 88.5 | **67.6** | 76.5 | **43.0** | 59.1 | **90.7** | 90.6 | **46.4** | 69.6 |

## 6.3 DISTILLATION FROM STRONGER PRETRAINED MODELS

In this section, we demonstrate that our method can be applied to distillation from strong pretrained models, including the ones that have larger model size and the ones are pretrained on larger-scale datasets.

**Scale up the pretrained model size.**    Larger pretrained models as teacher often hurts the student performance in knowledge distillation. As shown in Table 5, as the teacher model grows from ViT-Tiny, ViT-Small, ViT-Base to ViT-Large, the student model performance continues to decrease.

However, *ReMem* can effectively enable the distillation from larger pretrained models. As shown in Table 5, the student model performance improves consistently for teacher models with different sizes. Moreover, the improvement on larger teachers is more significant than that on small teachers, effectively enabling the student to learn more from larger pretrained models.

Table 5: Performance of the teacher and student with various teacher model sizes. We report the performance averaged over all 16 datasets.

| Method | ViT-Ti | | ViT-S | | ViT-B | | ViT-L | |
|---|---|---|---|---|---|---|---|---|
| | Student | Teacher | Student | Teacher | Student | Teacher | Student | Teacher |
| Vanilla Fine-tuning | 76.1 | 81.8 | 75.0 | 85.7 | 74.0 | 86.7 | 73.7 | 85.7 |
| *ReMem* | **77.9** | 81.9 | **78.4** | 84.8 | **78.3** | 85.7 | **78.5** | 84.6 |

**Scale up the pretraining dataset size.**    Models that are pretrained on larger-scale datasets may also hurt the student performance in knowledge distillation. As shown in Table 6 in the appendix, distilling from teachers that are pretrained on ImageNet-21k or ImageNet-21k with strong augmentations [8] consistently hurts the student performance compared to those pretrained on ImageNet-1k, although the teacher model performance improves significantly.

However, *ReMem* can effectively enable the distillation from models that are pretrained in larger scale. As shown in Table 6, the student model performance improves consistently for teacher models pretrained with larger datasets. Moreover, the improvement on teachers pretrained on a large scale is more significant, allowing the student to learn more from large-scale pretrained models.

Table 6: Performance of the teacher and student with various teacher pretrained datasets. We report the performance averaged over all 16 datasets.

| Method | ImageNet-1k | | ImageNet-21K | | ImageNet-21k (augreg) | |
|---|---|---|---|---|---|---|
| | Student | Teacher | Student | Teacher | Student | Teacher |
| Vanilla Fine-tuning | 75.8 | 84.5 | 73.2 | 86.2 | 74.0 | 86.7 |
| *ReMem* | **78.2** | 85.7 | **78.3** | 85.6 | **78.3** | 85.7 |

In Appendix B.1, we provide necessary ablation study of our method to show the individual effects of memory reweighting and SAM.

## 7    CONCLUSION AND DISCUSSION

In this paper, we explore simple and efficient methods to improve the knowledge distillation effectiveness of strong pretrained models. By observing that mutual information is essential for effective knowledge distillation, we propose to incorporate mutual information-aware optimization into fine-tuning. We alos localize the mutual information problem of pretrained models to mostly the top MLP blocks. We show the spontaneous expertness of these MLP blocks will greatly reduce the mutual information and impact the knowledge distillation effectiveness. We propose to further downweight these particular MLP blocks before employing strong pretrained models as teacher, which can significantly boost the effectiveness of the knowledge distillation in a widely variety of tasks and settings.

Our proposed method may also be suitable for use as a curriculum control method to gradually ramp up the difficulty of the teacher, which may be more effective or efficient than those conventional curriculum learning methods, for example, learning from small to large models sequentially, or from early to late checkpoints sequentially. However, due to limited computation resources, we would like to leave this exploration as a future work.

---

[8]Here we use the checkpoint from (Chen et al., 2021)

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

## A ADDITIONAL EXPERIMENT SETUP DETAILS

### A.1 DATASET SPECIFICATIONS

In table 7, we list the specifications of all datasets employed in this paper, including the number of classes, the sizes of the training and test set, as well as the metric used for evaluating the performance.

Table 7: Overview of image datasets. The datasets are train-test split based on TensorFlow Datasets defaults. When there is no default split available on TensorFlow Datasets, we split based on VTAB (Zhai et al., 2019), as denoted by $^\dagger$. We denote datasets that have a long-tail class distribution in the training set with $^*$. We define the metric used for each dataset following (Kornblith et al., 2018).

| Dataset | Classes | Size (train/test) | Accuracy metric |
|---|---|---|---|
| *Natural images* | | | |
| Food-101 (Bossard et al., 2014) | 101 | 75,750/25,250 | top-1 |
| CIFAR-100 (Krizhevsky, 2009) | 100 | 50,000/10,000 | top-1 |
| SUN397 (Xiao et al., 2010) (Kornblith et al. (2018) split) | 397 | 19,850/19,850 | top-1 |
| Stanford Cars (Krause et al., 2013) | 196 | 8,144/8,041 | top-1 |
| Caltech-101 (Fei-Fei et al., 2004) | 102 | 3,060/6,084 | mean per-class |
| Oxford 102 Flowers (Nilsback & Zisserman, 2008) | 102 | 2,040/6,149 | mean per-class |
| Pets (Parkhi et al., 2012) | 37 | 3,680/3,669 | top-1 |
| DTD (Cimpoi et al., 2014) | 47 | 1,880/1,880 | top-1 |
| SVHN (Netzer et al., 2011) | 10 | 73,257 / 26,032 | top-1 |
| SUN397$^*$ (Xiao et al., 2010) (TFDS default split) | 397 | 76,128/10,875 | mean per-class |
| ImageNet-LT$^*$ (Liu et al., 2019) | 1,000 | 115,846/20,000 | top-1 |
| iNaturalist 2017$^*$ (Van Horn et al., 2018) | 5,089 | 579,184/95,986 | mean per-class |
| *Medical images* | | | |
| Patch Camelyon (Veeling et al., 2018) | 2 | 262,144/32,768 | top-1 |
| Diabetic Retinopathy (Kaggle & EyePacs, 2015) | 5 | 35,126/10,906 | top-1 |
| *Sensing images* | | | |
| EuroSAT$^\dagger$ (Helber et al., 2017) | 10 | 21,600/5,400 | top-1 |
| Resisc45$^\dagger$ (Cheng et al., 2017) | 45 | 25,200/6,300 | top-1 |

### A.2 HYPERPARAMETER SETTINGS

In table 8, we list all hyperparameters employed in this paper as well as their choices.

### A.3 KNOWLEDGE DISTILLATION ALGORITHMS

**DIST.** When using DIST for knowledge distillation, we implement the loss function as follows.

$$L = (1 - \alpha)L_{cls} + \alpha L_{div}, \tag{7}$$

where

$$L_{div} = \beta L_{inter} + \gamma L_{intra}. \tag{8}$$

Here $L_{inter}$ and $L_{intra}$ are the inter-class loss and intra-class loss defined in Huang et al. (2022) respectively.

### A.4 PARAMETER-EFFICIENT FINE-TUNING METHODS

**LoRA.** For a pre-trained weight matrix $W$ in the transformer model, LoRA (Hu et al., 2021) constrains its update a low-rank decomposition

$$W + \Delta W = W + BA, \tag{9}$$

where $A$ and $B$ are too rank-deficient matrices. During fine-tuning, only the matrics $A$ and $B$ will be updated. Following the original paper, we peform random Gaussian initialization for $A$ and zero for

Table 8: Default hyperparameter setup for fine-tuning and distillation.

| Hyperparameters | Fine-tuning | Distillation |
|---|---|---|
| Training steps | {500, 700, 1000, 3000, 5000, 7000, 10000} | 2e4 |
| Optimizer | SGD | SGD |
| Learning rate scheduler | Cosine | Cosine |
| Peak learning rate | {0.005, 0.007, 0.01, 0.03, 0.05, 0.07, 0.1, 0.3, 0.5} | {0.05, 0.1, 0.5} (KD) {0.5, 0.7, 1.0} (DIST) |
| Warm-up steps | 500 | 500 |
| Batch size | 512 | 512 |
| Dropout | 0 | 0 |
| Weight decay | 0 | 1e-4 |
| MLP weight ($\alpha$) | {0.8, 0.9} | - |
| Attention weight ($\alpha$) | {0.8, 0.9} | - |
| SAM perturbation size ($\rho$) | {0.5, 0.05, 0.005} | - |
| LoRA rank ($r$) | 32 | - |
| LoRA scaling factor ($\alpha$) | 32 | - |
| Adapter reduction factor ($r$) | 16 | - |
| KD Temperature | - | {1, 2, 4} |
| KD loss weight ($\alpha$) | - | {0.1, 0.5, 0.9} |
| DIST inter loss weight ($\beta$) | - | 1.0 |
| DIST intra loss weight ($\gamma$) | - | 1.0 |
| Patient Mixup alpha | - | 0.8 |
| Patient training steps | - | 2e5 |

$B$. The output of $\Delta W$ will be further scaled by $\alpha/r$, where $\alpha$ is a constant scaling factor. We apply LoRA to only the Self-attention blocks, and specifically the query and value matrices following the original paper.

### A.5 ESTIMATE MUTUAL INFORMATION

We construct a decoder model to reconstruct the inputs from the features at the last layer to quantify the mutual information. We only compare the estimated mutual information of pretrained models with a same size of last layer features, otherwise it would not be fair. Our decoder model is based on a convolutional neural network, with detailed architecture shown in Table 9. We can employ this decoder architecture for all downstream datasets as we always resize the input images to $224 \times 224$.

Table 9: Architecture of the decoder network.

| Input: $14 \times 14$ feature maps |
|---|
| Bilinear Interpolation to $28 \times 28$ |
| $1 \times 1$ conv., stride=1, padding=0, output channels=128, BatchNorm+ReLU |
| Bilinear Interpolation to $56 \times 56$ |
| $3 \times 3$ conv., stride=1, padding=1, output channels=32, BatchNorm+ReLU |
| Bilinear Interpolation to $112 \times 112$ |
| $3 \times 3$ conv., stride=1, padding=1, output channels=12, BatchNorm+ReLU |
| Bilinear Interpolation to $224 \times 224$ |
| $3 \times 3$ conv., stride=1, padding=1, output channels=3, Sigmoid |

We train the decoder for 2000 updates to minimize the averaged binary cross-entropy reconstruction loss of all pixels. We employ the standard AdamW optimizer (Loshchilov & Hutter, 2017) for optimization, with hyper-parameters lr=0.001, betas=(0.9, 0.999), eps=1e-08, and weight decay=1e-4.

### A.6 COMPUTATION RESOURCES

Our experiments are conducted on several Nvidia RTX A5000 GPUs. A complete run of a single experiment including both teacher fine-tuning and student training, which typically takes about 7-8 hours to finish on a single GPU, when the teacher model is a ViT-base and the student model is a ResNet-18.

# B    ADDITIONAL EXPERIMENT RESULTS

## B.1    ABLATION STUDY

**Individual effects of memory reweighting and SAM.**    We toggle off memory reweighting or SAM in *ReMem* to explore their individual effects. As shown in Table 10, memory reweighting and SAM can both significantly boost the student performance, while the best performance can be achieved when the two are combined.

Table 10: Performance of the teacher and student with individual components or other variations of our method. ).

| Method | Student | Teacher |
|---|---|---|
| Baseline | 63.6 | 81.1 |
| Reweight MLP only | 65.1 | 81.3 |
| SAM only | 66.3 | 81.9 |
| Reweight MLP + SAM (*ReMem*) | **68.1** | 79.7 |

# C    ADDITIONAL ANALYSIS RESULTS

## C.1    DISTILLATION EFFECTIVENESS WITH BLOCK PRUNING

In Figure 4, we observed that pruning top MLP blocks can improve mutual information of the pretrained model without degrading its performance significantly. Here we show that this can indeed improve the distillation effectiveness of these pretrained models with top MLP blocks pruned. As shown in Figure 6, on multiple downstream datasets, we observed significant improvement of the student performance upon knowledge distillation when a proper number of MLP blocks of the pretrained model are pruned.

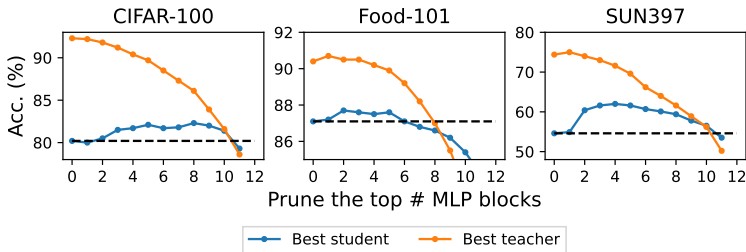

Figure 6: Performance of a fine-tuned pretrained model (teacher) versus that of a small model (student) distilled from it. Here the pretrained model is ViT-B pretrained on ImageNet-21k with strong data augmentation.

## C.2    VISUALIZATION OF NEURONS

We visualize the neurons in the top MLP blocks and show that they encode specific "skills" that are highly predictive of the class label, which echoes with the observation that top MLP blocks may have high expertness.

To visualize representative neurons, we conduct criticality analysis (Zhang et al., 2019; Maini et al., 2023) to reveal the dependence of model outputs on specific neurons. Specifically, on each example $x_i$, we record the relative change of the "[CLS]" token's embedding (*i.e.*, pre-logits representation) when setting an individual neuron $h$ to zero. We then use the maximum change over all examples to denote the criticality of a specific neuron to the model output, namely $\sigma(h) = \max_{i \in [N]} \|F_i^{h \leftarrow 0} - F_i\| / \|F_i\|$, where $F_i$ denotes the "[CLS]" embedding of the $i$-th input example, and $N$ is the total number of input examples.

In Figure 7, we visualize 10 neurons in the top MLP block that influence the model outputs the most based on the criticality analysis. One can observe that these neurons encode complete and concrete object features. For each neuron, we also show the downstream training example that is most influenced by it [9]. One can observe that the input example can almost be exactly matched by the visualization of the corresponding neuron . This implies that each of these neurons may encode the knowledge that is sufficient to recognize the a similar class of examples in the downstream dataset on its own. In contrast, for bottom MLP blocks and for weak models, we rarely observe any concrete objects in the knowledge encoded by their neurons, as shown in Figures 8, 9, 10, 11, 12, 13. Such phenomena are also observed in different downstream datasets (Figures 14, 15).

These evidences suggest that top MLP blocks may indeed behave as MoE, where each neuron singly or a few neurons collectively act as an expert, and the partition of the experts may largely based on the class label.

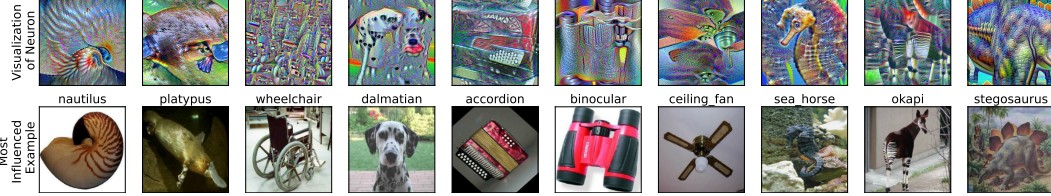

Figure 7: (Top): Visualization of the most critical neurons in the 10-th MLP block. Here the model is a ViT-B model pretrained on ImageNet-21k. (Bottom): Image examples that are most influenced by the corresponding neurons in the downstream dataset (Caltech-101).

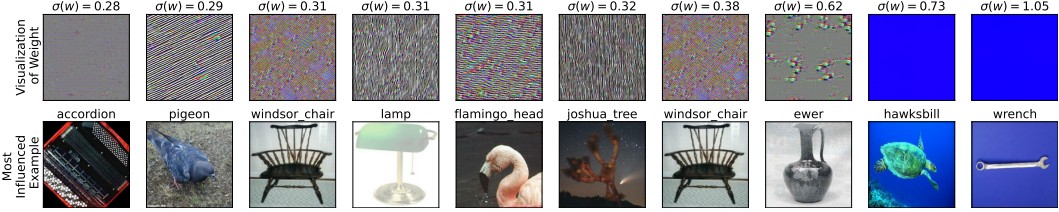

Figure 8: (Top) Visualization of the most critical neurons in the 0-th MLP block. Here the model is a pretrained but not fine-tuned ViT-Base model. (Bottom) Image examples that are most influenced by the corresponding neurons. Here the downstream dataset is Caltech-101.

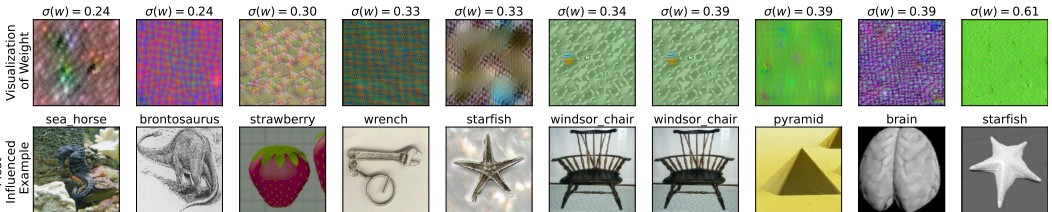

Figure 9: (Top) Visualization of the most critical neurons in the 3-th MLP block. Here the model is a pretrained but not fine-tuned ViT-Base model. (Bottom) Image examples that are most influenced by the corresponding neurons. Here the downstream dataset is Caltech-101.

---

[9]An example is influenced by a neuron if its model output changes significantly when zeroing out this neuron.

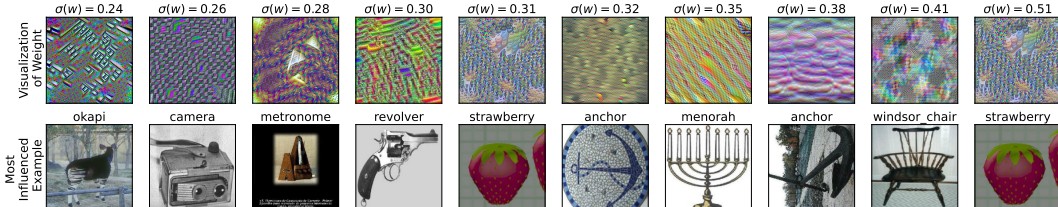

Figure 10: (Top) Visualization of the most critical neurons in the 5-th MLP block. Here the model is a pretrained but not fine-tuned ViT-Base model. (Bottom) Image examples that are most influenced by the corresponding neurons. Here the downstream dataset is Caltech-101.

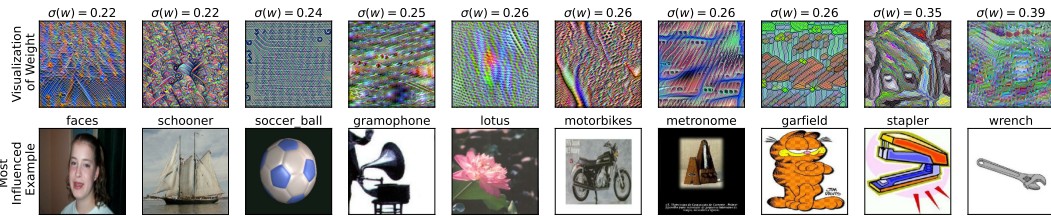

Figure 11: (Top) Visualization of the most critical neurons in the 7-th MLP block. Here the model is a pretrained but not fine-tuned ViT-Base model. (Bottom) Image examples that are most influenced by the corresponding neurons. Here the downstream dataset is Caltech-101.

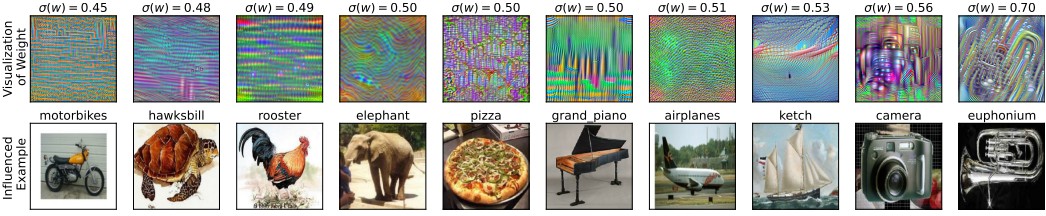

Figure 12: (Top) Visualization of the most critical neurons in the 10-th MLP block. Here the model is a pretrained but not fine-tuned ViT-Tiny model. (Bottom) Image examples that are most influenced by the corresponding neurons. Here the downstream dataset is Caltech-101.

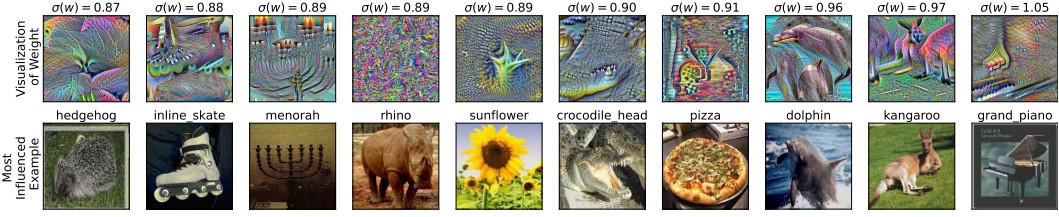

Figure 13: (Top) Visualization of the most critical neurons in the 10-th MLP block. Here the model is a pretrained but not fine-tuned ViT-Small model. (Bottom) Image examples that are most influenced by the corresponding neurons. Here the downstream dataset is Caltech-101.

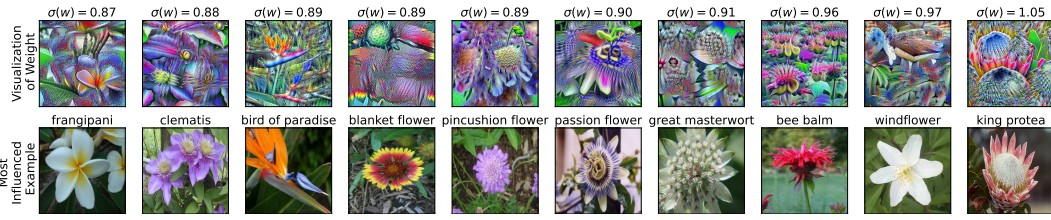

Figure 14: (Top) Visualization of the most critical neurons in the 10-th MLP block. Here the model is a pretrained but not fine-tuned ViT-Base model. (Bottom) Image examples that are most influenced by the corresponding neurons. Here the downstream dataset is Flowers-102.

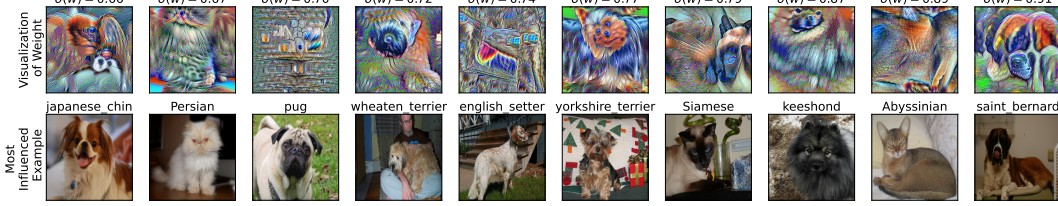

Figure 15: (Top) Visualization of the most critical neurons in the 10-th MLP block. Here the model is a pretrained but not fine-tuned ViT-Base model. (Bottom) Image examples that are most influenced by the corresponding neurons. Here the downstream dataset is Pet.

