# OpenReview forum: "How To Be A Good Teacher? Process Strong Pretrained Models For Effective Knowledge Distillation"
_ICLR.cc/2025/Conference — Submitted to ICLR 2025_

### Official Review · Reviewer_Ecsz · 2024-11-03

**Soundness:** 2
**Presentation:** 3
**Contribution:** 3
**Rating:** 5
**Confidence:** 4

**Summary:**

This paper presents methods to enhance the knowledge distillation process, specifically when using strong pretrained models as teachers for smaller, task-specific models. The authors propose mutual information-aware optimization and MLP block reweighting (which they call ReMem) to improve the distillation process. The work emphasizes the critical role of mutual information between the teacher and student models in effective knowledge transfer, aiming to reduce information loss during distillation. The paper also provided good insights on the Transformer architecture, and how different components operate during knowledge distillation. The method demonstrates effectiveness across varying model sizes, dataset scales, and architecture types, including efficient models like ResNet18, MobileNetV2 and EfficientNetV2. The authors show that with reweighting based fine-tuning of teacher models, the student model's performance can be significantly improved post distillation.

**Strengths:**

1. This work addresses a key limitation in distillation from large models, where standard approaches often struggle. By considering mutual information in the fine-tuning objective, the authors mitigate the challenges of larger teacher models while distilling knowledge to smaller models. The problem statement and the mitigation method are novel and relevant.

2. The proposed methodology is proven effective across varying model sizes and datasets, which makes the methodology more adaptable to different use cases.

3. The critical observations made in this paper are particularly useful to understand how knowledge distillation (KD) works. Not many works have been done on KD interpretation. Therefore, any proven conjecture on the effectiveness of KD can be beneficial for the research community to have a better grasp on this field.

**Weaknesses:**

The experimental part lacks rigour and could be improved further. Following are my suggestions -

1. Highlight ablation results with ReZero instead of reweighting.
2. It is important to understand the benefit of SAM over other methods like MINE. The authors should consider experimenting with ReMem with other optimization methods like MINE. Or at least some theoretical results should be provided to analytically justify the selection of SAM.
3. There exists distillation methods that do not require teacher fine-tuning. For instance - Sengupta et al., 2023 [1] show that a competitive loss between teacher and student can enhance the effectiveness of distillation. The authors should consider countering these methods, either analytically or empirically.
4. See the following questions.

**References**

[1] A Good Learner can Teach Better: Teacher-Student Collaborative Knowledge Distillation, ICLR'2024.

**Questions:**

1. When the downstream dataset is small,the above mutual information-aware optimization may not be effective. Why?
2. Why does ReMem reduces teacher performance in table 2 and 3?
3. What is patient in Table 3?
4. Where is the proof of proposition 5.4?
5. What happens to expertisness after applying ReMem?

---

### Official Review · Reviewer_xFbt · 2024-11-04

**Soundness:** 2
**Presentation:** 2
**Contribution:** 2
**Rating:** 6
**Confidence:** 4

**Summary:**

In this paper, the author examines knowledge distillation in large pretrained models, proposing that inefficiency may stem from the large model itself, as low mutual information can hinder effective knowledge transfer to the student model. To address this, they propose a weight redistribution method to mitigate this issue and use it with an existing SAM method.

**Strengths:**

First of all, I do like the angle of teacher-side improvement for the knowledge distillation problem. The proposed method is easy to adopt and quite intuitive. It also provides a certain level of insight into how knowledge is preserved in the pre-trained models. The structure and the organization of the paper also look good to me.

**Weaknesses:**

My concerns mainly arise from the following aspects:

The findings/insights regarding the depletion of mutual information in the pretrained model may require more supporting evidence. The author defines "expertiseness" using a bipartite graph, framing it as a bipartite problem. While this formulation helps bridge expertiseness with mutual information, the connection between expertiseness and the MLP layers is weak, raising questions about the generalizability of this finding. Additionally, the author focuses on specific models (ViT) to demonstrate this phenomenon, which may not be applicable to pretrained models in general, or at least, the generalization of these findings has not been demonstrated.

Another concern is whether this weight redistribution might interfere with other domain knowledge (beyond what the author aims to transfer) that the teacher model has already learned.

**Questions:**

1. In any of your experiment sections, could you demonstrate the level of expertise both before and after your optimization, and perhaps show whether such an alteration indeed enhances knowledge transfer?

2. How generalizable is this approach across different pretrained models?

3. What criteria do you use to select which MLP layers should be trained, especially when working with large-scale models?

4. How do you ensure that the model’s performance is not compromised by the weight redistribution changes? Could it fail in other tasks, and are there any guarantees to prevent this?

---

### Official Review · Reviewer_tsUj · 2024-11-05

**Soundness:** 2
**Presentation:** 3
**Contribution:** 2
**Rating:** 3
**Confidence:** 5

**Summary:**

This paper focuses on how to improve the knowledge transfer of a powerful pretrained model in the knowledge distillation process. The main contribution of this paper lies in developing two methods to optimize the distillation effect of the model.

**Strengths:**

By preprocessing powerful pre-trained models prior to distillation (such as mutual information optimization and MLP block reweighting), the efficiency of large models in knowledge transfer is addressed, enabling student models to learn better from teacher models. For example, mutual information optimization through Sharpness-Aware Minimization, while MLP block reweighting is a heuristic method that is relatively easy to implement and can be directly integrated into the existing distillation process.

**Weaknesses:**

Although the paper validates on models such as ViT (Vision Transformer), the applicability of this approach to other types of pre-trained models, such as large language models, is unclear. The architectural characteristics of different models may lead to different effects of mutual information optimization and MLP reweighting.
Although mutual information awareness optimization can theoretically help improve distillation performance, its effect is limited when the data sets are small or unbalanced. In the paper, the authors acknowledge this situation and propose a heuristic MLP reweighting method as a supplement. However, the effectiveness of this approach depends on specific data and models, and may not be suitable for all scenarios.
Mutual information optimization and MLP reweighting, while effective, add complexity to the model. This can make the model more difficult to interpret and debug.
Mutual information optimization (such as SAM) and MLP reweighting methods both introduce additional hyperparameters (such as disturbance size and weight factor of MLP block). Although the setting of these parameters has great impact on the results, the paper does not provide a hyperparameter tuning strategy, which may increase the complexity of model tuning.
Although they demonstrated the overall results of their approach, there is a lack of ablation experiments for the various components of mutual information optimization and MLP reweighting.

**Questions:**

What's the effectiveness of the proposed heuristic MLP reweighting scheme?

What's the  computational complexity of the proposed method?

How do you tune the hyperparameters?

There is a lack of ablation experiments.

---

### Meta-Review · Area_Chair_exqj · 2024-12-20

**Metareview:**

Summary: This paper pointed out that preprocessing stong pretrained models can help with knowledge distillation. Thie work further proposed to incorporate mutual information-aware optimization for fine-tuning the strong teacher models.

Strengths: (1) The idea of preprocessing large strong pretrained models to improve knowledge distillation is novel and interesting. (2) The proposed method is effective across different model sizes and datasets

Weaknesses: The main weakness is that the experiments are not comprehensive and need more supporting evidence, including the depletion of mutual information, models beyond ViT, ablation results with ReZero, experiments with other optimization methods, etc.

This paper received ratings as 6, 5, 3. The authors did not respond to reviewers' comments. The reviewer who gave the rating as 6 is not fighting for accepting the paper.

**Additional Comments On Reviewer Discussion:**

The authors did not respond to reviewers' comments. The reviewer who gave the rating as 6 is not fighting for accepting the paper.

---

### Decision · Program_Chairs · 2025-01-22

Reject